# Genetic diagnosis of Mendelian disorders via RNA sequencing

Laura S. Kremer[1,2,*], Daniel M. Bader[3,4,*], Christian Mertes[3], Robert Kopajtich[1,2], Garwin Pichler[5], Arcangela Iuso[1,2], Tobias B. Haack[1,2,†], Elisabeth Graf[1,2], Thomas Schwarzmayr[1,2], Caterina Terrile[1], Eliška Koňaříková[1,2], Birgit Repp[1,2], Gabi Kastenmüller[6], Jerzy Adamski[7], Peter Lichtner[1], Christoph Leonhardt[8], Benoit Funalot[9], Alice Donati[10], Valeria Tiranti[11], Anne Lombes[12,13,14], Claude Jardel[12,15], Dieter Gläser[16], Robert W. Taylor[17], Daniele Ghezzi[11], Johannes A. Mayr[18], Agnes Rötig[9], Peter Freisinger[19], Felix Distelmaier[20], Tim M. Strom[1,2], Thomas Meitinger[1,2], Julien Gagneur[3,4] & Holger Prokisch[1,2]

Across a variety of Mendelian disorders, ∼50–75% of patients do not receive a genetic diagnosis by exome sequencing indicating disease-causing variants in non-coding regions. Although genome sequencing in principle reveals all genetic variants, their sizeable number and poorer annotation make prioritization challenging. Here, we demonstrate the power of transcriptome sequencing to molecularly diagnose 10% (5 of 48) of mitochondriopathy patients and identify candidate genes for the remainder. We find a median of one aberrantly expressed gene, five aberrant splicing events and six mono-allelically expressed rare variants in patient-derived fibroblasts and establish disease-causing roles for each kind. Private exons often arise from cryptic splice sites providing an important clue for variant prioritization. One such event is found in the complex I assembly factor TIMMDC1 establishing a novel disease-associated gene. In conclusion, our study expands the diagnostic tools for detecting non-exonic variants and provides examples of intronic loss-of-function variants with pathological relevance.

[1] Institute of Human Genetics, Helmholtz Zentrum München, 85764 Neuherberg, Germany. [2] Institute of Human Genetics, Klinikum rechts der Isar, Technische Universität München, 81675 München, Germany. [3] Department of Informatics, Technische Universität München, 85748 Garching, Germany. [4] Quantitative Biosciences Munich, Gene Center, Department of Biochemistry, Ludwig Maximilian Universität München, 81377 München, Germany. [5] Department of Proteomics and Signal Transduction, Max-Planck Institute of Biochemistry, 82152 Martinsried, Germany. [6] Institute of Bioinformatics and Systems Biology, Helmholtz Zentrum München, 85764 Neuherberg, Germany. [7] Institute of Experimental Genetics, Genome Analysis Center, Helmholtz Zentrum München, German Research Center for Environmental Health, 85764 Neuherberg, Germany. [8] Neuropädiatrie, Neonatologie, 78050 Villingen-Schwenningen, Germany. [9] INSERM U1163, Université Paris Descartes—Sorbonne Paris Cité, Institut Imagine, 75015 Paris, France. [10] Metabolic Unit, A. Meyer Children's Hospital, 50139 Florence, Italy. [11] Unit of Molecular Neurogenetics, Foundation IRCCS (Istituto di Ricovero e Cura a Carettere Scientifico) Neurological Institute 'Carlo Besta', 20126 Milan, Italy. [12] Inserm UMR 1016, Institut Cochin, 75014 Paris, France. [13] CNRS UMR 8104, Institut Cochin, 75014 Paris, France. [14] Université Paris V René Descartes, Institut Cochin, 75014 Paris, France. [15] AP/HP, GHU Pitié-Salpêtrière, Service de Biochimie Métabolique, 75013 Paris, France. [16] Genetikum, Genetic Counseling and Diagnostics, 89231 Neu-Ulm, Germany. [17] Wellcome Centre for Mitochondrial Research, Institute of Neuroscience, Newcastle University, Newcastle upon Tyne NE2 4HH, UK. [18] Department of Pediatrics, Paracelsus Medical University, A-5020 Salzburg, Austria. [19] Department of Pediatrics, Klinikum Reutlingen, 72764 Reutlingen, Germany. [20] Department of General Pediatrics, Neonatology and Pediatric Cardiology, University Children's Hospital, Heinrich-Heine-University Düsseldorf, 40225 Düsseldorf, Germany. † Present address: Institute of Medical Genetics and Applied Genomics, University of Tübingen, 72076 Tübingen, Germany. * These authors contributed equally to this work. Correspondence and requests for materials should be addressed to J.G. (email: gagneur@in.tum.de) or to H.P. (email: prokisch@helmholtz-muenchen.de).

Despite the revolutionizing impact of whole-exome sequencing (WES) on the molecular genetics of Mendelian disorders, ~50–75% of the patients do not receive a genetic diagnosis after WES[1]. The disease-causing variants might be detected by WES but remain as variants of unknown significance (VUS, Methods section) or they are missed due to the inability to prioritize them. Many of these VUS are synonymous or non-coding variants that may affect RNA abundance or isoform but cannot be prioritized due to the poor understanding of regulatory sequence to date compared to coding sequence. Furthermore, WES covers only the 2% exonic regions of the genome. Accordingly, it is mostly blind to regulatory variants in non-coding regions that could affect RNA sequence and abundance. While the limitation of genome coverage is overcome by whole genome sequencing (WGS), prioritization and interpretation of variants identified by WGS is in turn limited by their amount[2–4].

With RNA sequencing (RNA-seq), limitations of the sole genetic information can be complemented by directly probing variations in RNA abundance and in RNA sequence, including allele-specific expression and splice isoforms. At least three extreme situations can be directly interpreted to prioritize candidate disease-causing genes for a rare disorder. First, the expression level of a gene can lie outside its physiological range. Genes with expression outside their physical range can be identified as expression outliers, often using a stringent cutoff on expression variations, for instance using the Z-score[5] or statistics at the level of whole gene sets[6,7]. The genetic causes of such aberrant expression includes rare variants in the promoter[8] and enhancer but also in coding or intronic regions[5]. Second, RNA-seq can reveal extreme cases of allele-specific expression (mono-allelic expression (MAE)), whereby one allele is silenced, leaving only the other allele expressed. When assuming a recessive mode of inheritance, genes with a single heterozygous rare coding variant identified by WES or WGS analysis are not prioritized. However, MAE of such variants fits the recessive mode of inheritance assumption. Detection of MAE can thus help re-prioritizing heterozygous rare variants. Reasons for MAE can be genetic. A pilot study validated compound heterozygous variants within one gene as cause of TAR syndrome, where one allele is deleted and the other harbours a non-coding variant that reduces expression[9]. MAE can also have epigenetic causes such as X-chromosome inactivation or imprinting on autosomal genes, possibly by random choice[10,11]. Third, splicing of a gene can be affected. Aberrant splicing has long been recognized as a major cause of Mendelian disorders (reviewed in refs 12–14). However, the prediction of splicing defects from genetic sequence is difficult because splicing involves a complex set of cis-regulatory elements that are not yet fully understood. Some of them can be deeply located in intronic sequences[15] and are thus not covered by WES. Hence, direct probing of splice isoforms by RNA-seq is important, and has led to the discovery of multiple splicing defects based on single gene studies: skipping of multiple exons (exon 45–55)[16] and creation of a new exon by a deep intronic variant in DMD[17], intron retention in LMNA caused by a 5′ splice site variant[18], and skipping of exon 7 in SMN1 caused by a variant in a splicing factor binding site[19]. Altogether, RNA-seq promises to be an important complementary tool to facilitate molecular diagnosis of rare genetic disorders. However, no systematic study to date has been conducted to assess its power.

We considered investigating the power of RNA-sequencing for molecular diagnosis with a panel of patients diagnosed with a mitochondrial disorder for three reasons. First, mitochondrial diseases collectively represent one of the most frequent inborn errors of metabolism affecting 2 in 10,000 individuals[20]. Second,

the broad range of unspecific clinical symptoms and the genetic diversity in mitochondrial diseases makes molecular diagnosis difficult and WES often results in VUS. As a consequence of the bi-genomic control of the energy-generating oxidative phosphorylation (OXPHOS) system, mitochondrial diseases may result from pathogenic mutations of the mitochondrial DNA or nuclear genome. More than 1,500 different nuclear genes encode mitochondrial proteins[21] and causal defects have been identified in ~300 genes and presumably more additional disease-associated genes still awaiting identification[22]. Third, since the diagnosis often relies on biochemical testing of a tissue sample, fibroblast cell lines are usually available from those patients. Moreover, for many patients, the disease mechanisms can be assayed in epidermal fibroblast cell lines even though the disease may manifest in different tissues[23]. This allows rapid demonstration of the necessary and sufficient role of candidate variants by perturbation and complementation assays[24]. This also indicates that disease-causing expression defects, if any, should be detectable in these cell lines.

Here, we establish an analysis pipeline to systematically detect RNA defects to complement genome-based molecular diagnosis. We find a median of one aberrantly expressed gene, five aberrant splicing events and six mono-allelically expressed rare variants in patient-derived fibroblasts and establish disease-causing roles for each kind. This leads to the molecular diagnosis of 10% of undiagnosed mitochondriopathy patients (5 out of 48) and yields candidate genes for 36 other patients. Systematic analysis of private exons shows that these often occur at locations where splicing is detectable at basal level in the population, providing an important clue for variant prioritization. Altogether, our study expands the tools for interpreting non-exonic variants and accelerating the genetic diagnosis of rare disorders.

## Results

**RNA sequencing on patient-derived fibroblasts.** We performed RNA-seq on 105 fibroblast cell lines from patients with a suspected mitochondrial disease including 48 patients for which WES based variant prioritization did not yield a genetic diagnosis (Fig. 1, Methods section, Supplementary Table 1). After discarding lowly expressed genes, RNA-seq identified 12,680 transcribed genes (at least 10 reads in 5% of all samples, Methods section, Supplementary Data 1). We systematically prioritized genes with the following three strategies: (i) genes with aberrant expression level[6–8], (ii) genes with aberrant splicing[17,25] and (iii) MAE of rare variants[9] (Fig. 1) to estimate their disease association. All strategies are based on the comparison of one patient against the rest. We assumed the causal defects to differ between patients, which is reasonable for mitochondrial disorders with a diversity of ~300 known disease-causing genes (Supplementary Data 2). Therefore, the patients serve as good controls for each other.

**Aberrant expression.** Once normalized for technical biases, sex and biopsy site (Supplementary Method 1 and Supplementary Fig. 1), the samples typically presented few aberrantly expressed genes (median of 1, 90% of the samples with <10, Fig. 2a, Supplementary Data 3) with a large effect ($|Z\text{-score}| > 3$) and significant differential expression (Hochberg adjusted $P$ value < 0.05). Among the most aberrantly expressed genes across all samples, we found 2 genes encoding mitochondrial proteins, MGST1 (one case) and TIMMDC1 (two cases) to be significantly down-regulated (Fig. 2b–d and Supplementary Fig. 2). For both genes, WES did not identify any variants in the respective patients, no variant is reported to be

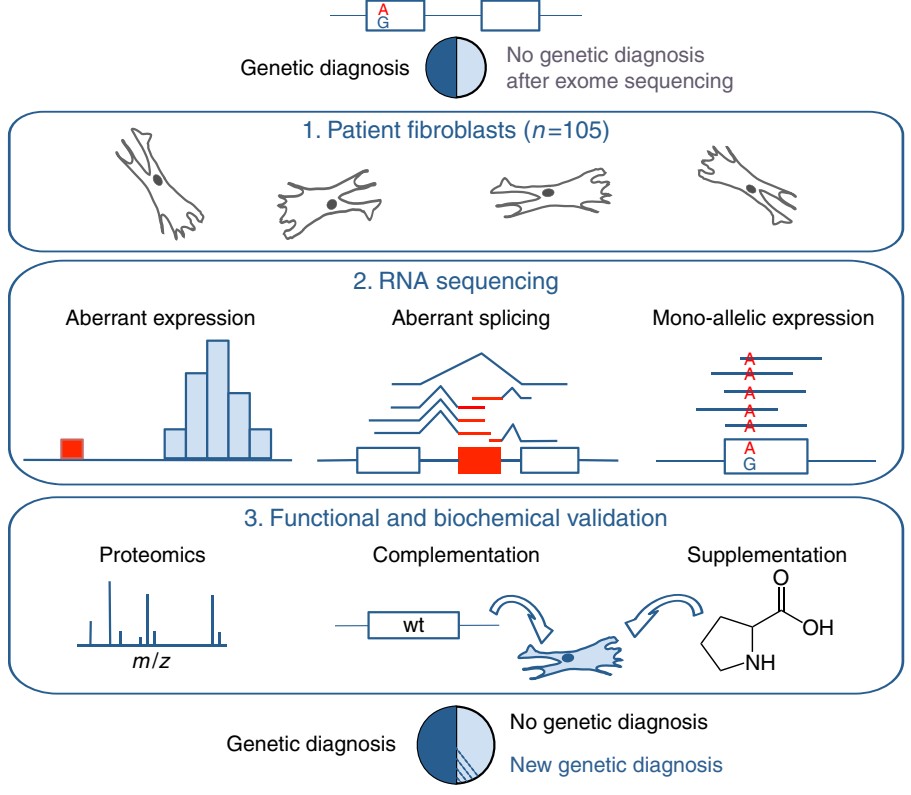

**Figure 1 | Strategy for genetic diagnosis using RNA-seq.** The approach we followed started with RNA-seq of fibroblasts from unsolved WES patients. Three strategies to facilitate diagnosis were pursued: Detection of aberrant expression (for example, depletion), aberrant splicing (for example, exon creation) and MAE of the alternative allele (for example, A as alternative allele). Candidates were validated by proteomic measurements, lentiviral transduction of the wild-type (wt) allele or, in particular cases, by specific metabolic supplementation.

disease-associated and no case of potential bi-allelic rare variant is listed in our in-house database comprising more than 1,200 whole-exomes from mitochondrial patients and more than 15,000 whole-exomes available to us from different ongoing research projects. To evaluate the consequences of diminished RNA expression at the protein level, we performed quantitative proteomics in a total of 31 fibroblast cell lines (including these three patients, and further 17 undiagnosed and 11 diagnosed patients, Supplementary Table 1, Supplementary Data 4 and 5, Supplementary Methods 2–5) from a second aliquot of cells taken at the same time as the RNA-seq aliquot. Normalized RNA and protein expression levels showed a median rank correlation of 0.59, comparable to what has been previously reported[26,27] (Supplementary Fig. 3). Patient #73804 showed $\sim 2\%$ of control MGST1 level whilst the lack of detection of TIMMDC1 in both patients (#35791 and #66744) confirmed an even stronger effect on protein expression, indicating loss of function (Fig. 2e and Supplementary Fig. 4). MGST1, a microsomal glutathione S-transferase, is involved in the oxidative stress defense[28]. Consequently, the loss of expression of MGST1 is not only a likely cause of the disease of this patient, who suffers from an infantile-onset neurodegenerative disorder similar to a recently published case with another defect in the reactive oxygen species defense system (Supplementary Fig. 4, Supplementary Note 1)[29], but also suggests a treatment with antioxidants. Both TIMMDC1 patients presented with muscular hypotonia, developmental delay and neurological deterioration, which led to death in the first 3 years of life (Supplementary Note 1). Consistent with the described function of TIMMDC1 as a respiratory chain complex I assembly factor[30,31], we found isolated complex I

deficiency in muscle (Supplementary Fig. 2), and globally decreased levels of complex I subunits in fibroblasts by quantitative proteomics (Fig. 2e and Supplementary Fig. 2) and western blot (Fig. 2f, Supplementary Fig. 10). Re-expression of TIMMDC1 in these cells increased complex I subunit levels (Fig. 2f). These results not only validate TIMMDC1-deficiency as disease causing but also provide compelling evidence for an important function of TIMMDC1 in complex I assembly.

**Aberrant splicing**. We identified aberrant splicing events by testing for differential splicing in each patient against the others, using an annotation-free algorithm able to detect splice sites *de novo* (Methods section, median of 5 abnormal events per sample, 90% with $<16$, Fig. 3a). Among the 175 aberrant spliced genes detected in the undiagnosed patients, the most abundant events were, apart from differential expression of isoforms, exon skipping followed by the creation of new exons (Fig. 3b). Two genes encoding mitochondrial proteins, TIMMDC1 and CLPP, which were among the 20 most significant genes, caught our attention (Supplementary Data 6). Out of 136 exon-junction reads overlapping the acceptor site of CLPP exon 6 for patient #58955, 82 (per cent spliced in ref. 32, $\Psi = 60\%$) skipped exon 5, and 14 ($\Psi = 10\%$) showed a 3′-truncated exon 5, in striking contrast to other samples (Fig. 3c). The likely genetic cause of these two splice defects is a rare homozygous variant in exon 5 of CLPP affecting the last nucleotide of exon 5 (c.661G>A, p.Glu221Lys $1.2 \times 10^{-5}$ frequency in the ExAC database[33], Supplementary Fig. 5). Both detected splice defects result in truncated CLPP and western blots corroborated the complete loss

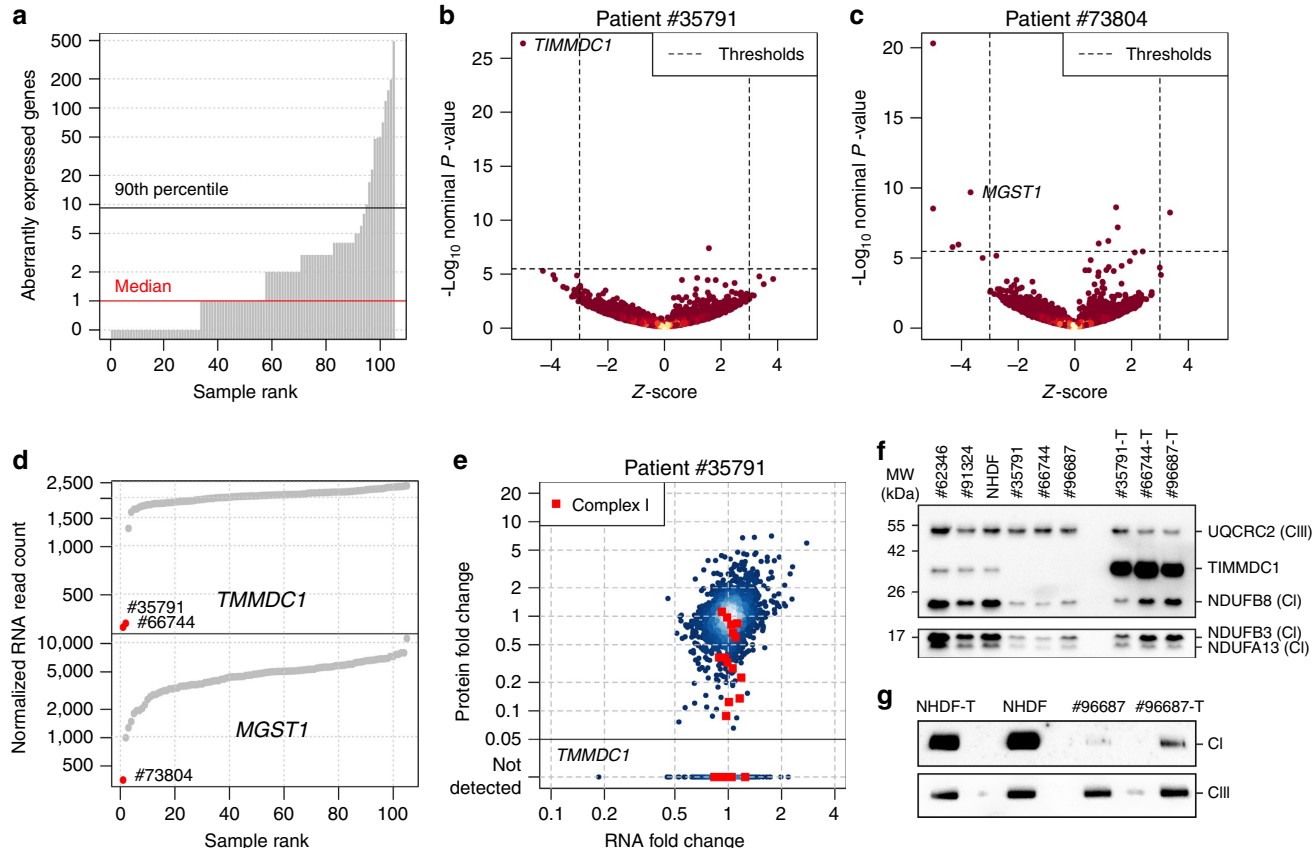

**Figure 2 | RNA aberrant expression detection and validation.** (**a**) Aberrantly expressed genes (Hochberg corrected $P$ value $< 0.05$ and $|Z\text{-score}| > 3$) for each patient fibroblasts. (**b**) Gene-wise RNA expression volcano plot of nominal $P$ values ($-\log_{10} P$ value) against $Z$-scores of the patient #35791 compared against all other fibroblasts. $Z$-scores with absolute value $> 5$ are plotted at $\pm 5$, respectively. (**c**) Same as (**b**) for patient #73804. (**d**) Sample-wise RNA expression is ranked for the genes $TIMMDC1$ (top) and $MGST1$ (bottom). Samples with aberrant expression for the corresponding gene are highlighted in red (#35791, #66744, and #73804). (**e**) Gene-wise comparison of RNA and protein fold changes of patient #35791 compared to the average across the fibroblast cell lines of all other patients. Subunits of the mitochondrial respiratory chain complex I are highlighted (red squares). Reliably detected proteins that were not detected in this sample are shown separately with their corresponding RNA fold changes (points below solid horizontal line). (**f**) Western blot of TIMMDC1, NDUFA13, NDUFB3 and NDUFB8 protein in three fibroblast cell lines without (#62346, #91324, NHDF) and three with a variant in $TIMMDC1$ (#35791, #66744 and #96687), and fibroblasts re-expressing $TIMMDC1$ ('-T') (#35791-T, #66744-T and #96687-T). UQCRC2 was used as loading control. CI, complex I subunit; CIII, complex III subunit; MW, molecular weight. (**g**) Blue native PAGE blot of the control fibroblasts re-expressing $TIMMDC1$ (NHDF-T), the control fibroblasts (NHDF), patient fibroblasts (#96687) and patient fibroblast re-expressing $TIMMDC1$ (#96687-T). Immunodecoration for complex I and complex III was performed using NDUFB8 and UQCRC2 antibodies, respectively. CI, complex I subunit; CIII, complex III subunit.

of full-length CLPP (Supplementary Figs 5 and 11). Our WES variant filtering reported this variant as a VUS and classified *CLPP* as one among 30 other potentially bi-allelic affected candidate genes (Supplementary Data 7). Since the variant was of unknown significance, the patient remained without genetic diagnosis. The loss of function found by RNA-seq and confirmed by western blotting now highlights clinical relevance of the variant within *CLPP*. *CLPP* encodes a mitochondrial ATP-dependent endopeptidase[34] and CLPP-deficiency causes Perrault syndrome[35,36] (OMIM #601119) which is overlapping with the clinical presentation of the patient investigated here including microcephaly, deafness and severe psychomotor retardation (Supplementary Note 1). Moreover, a study recently showed that Clpp $-/-$ mice are deficient for complex IV expression[37], in line with complex IV deficiency of this patient (Supplementary Fig. 5).

Split read distribution indicated that both TIMMDC1-deficient patients expressed almost exclusively a *TIMMDC1* isoform with a new exon in intron 5 (Fig. 3d). This new exon introduces a frameshift yielding a premature stop codon (p.Gly199_Thr200ins5*, Fig. 3e). Moreover, this new exon contained a rare variant (c.596 + 2146A > G) not listed in the 1,000 Genomes Project[2,3] (validated by Sanger sequencing, Supplementary Fig. 2, Supplementary Method 6). WGS demonstrated that this variant is homozygous in both patients (Fig. 3e, Supplementary Method 7), the only rare variant in this intron and close to the splice site (+6 of the new exon). We could not identify any rare variant in the promoter region or in any intron–exon boundary of *TIMMDC1*. Additionally, when testing six prediction tools for splicing events, this deep intronic rare variant is predicted by SpliceAid2 (ref. 38) to create multiple binding sites for splice enhancers. Together with the correctly predicted new acceptor and donor sites by SplicePort[39] (Feature generation algorithm score 0.112 and 1.308, respectively) this emphasizes the influence of this variant in the creation of the new exon. Besides, the four other tools predicted no significant change in splicing[40–43]. We further discovered an additional family in our in-house WGS database (consisting of 36 patients with a suspected mitochondrial disorder and 232 further patients with

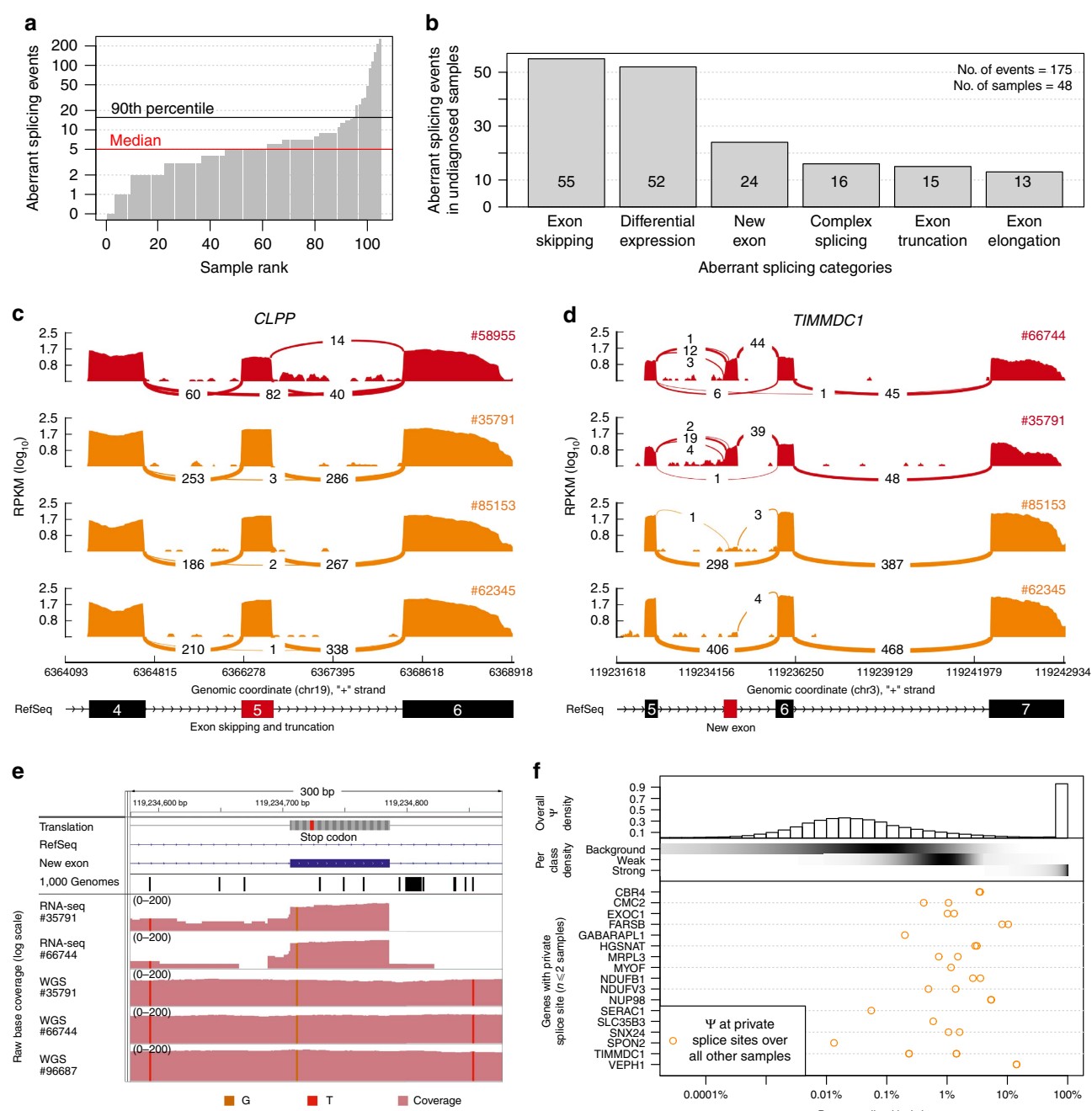

**Figure 3 | Aberrant splicing detection and quantification. (a)** Aberrant splicing events (Hochberg corrected $P$ value $< 0.05$) for all fibroblasts.
**(b)** Aberrant splicing events ($n = 175$) in undiagnosed patients ($n = 48$) grouped by their splicing category after manual inspection. **(c)** *CLPP* Sashimi plot of exon skipping and truncation events in *CLPP*-affected and *CLPP*-unaffected fibroblasts (red and orange, respectively). The RNA coverage is given as the $\log_{10}$ RPKM-value and the number of split reads spanning the given intron is indicated on the exon-connecting lines. At the bottom the gene model of the RefSeq annotation is depicted and the aberrantly spliced exon is coloured in red. **(d)** Same as in **c** for *TIMMDC1*. At the bottom the newly created exon is depicted in red within the RefSeq annotation track. **(e)** Coverage tracks (light red) for patients #35791, #66744, and #91324 based on RNA and WGS. For patient #91324 only WGS is available. The homozygous SNV c.596 + 2146A > G is present in all coverage tracks (vertical orange bar). The top tracks show the genomic annotation: genomic position on chromosome 3, DNA sequence, amino acid translation (grey, stop codon in red), the RefSeq gene model (blue line), the predominant additional exon of *TIMMDC1* (blue rectangle) and the SNV annotation of the 1000 Genomes Project (each black bar represents one variant). **(f)** Per cent spliced in ($\Psi$) distribution for different splicing classes and genes. Top: histogram of the genome-wide distribution of the 3' and 5' $\Psi$-values based on all reads over all samples. Middle: The shaded horizontal bars represent the densities (black for high density) of the background, weak and strong splicing class, respectively (Methods section). Bottom: $\Psi$-values of the predominant donor and acceptor splice sites of genes with private splice sites (that is, found predominant in at most two samples) computed over all other samples.

unrelated diseases) carrying the same homozygous intronic variant. In this family three affected siblings presented with similar clinical symptoms although without a diagnosis of a mitochondrial disorder

(Fig. 3e, Supplementary Fig. 2). Two siblings died before the age of 10. A younger brother (#96687), now 6 years of age, presented with muscle hypotonia, failure to thrive and neurological impairment

(Supplementary Note 1), similar to the patients described above. Western blot analysis confirmed TIMMDC1-deficiency (Fig. 2f, Supplementary Fig. 10) and impaired complex I assembly, which was restored after re-expression of *TIMMDC1* (Fig. 2g, Supplementary Fig. 10). The discovery of the same intronic *TIMMDC1* variant in three unrelated families from three different ethnicities provides convincing evidence on the causality of this variant for the TIMMDC1 loss-of-function.

In almost all non-TIMMDC1-deficiency samples, we noticed a few split reads supporting inclusion of the new exon (Fig. 3d), consistent with an earlier report that many cryptic splice sites are not entirely repressed but active at low levels[44]. We set out to quantify this phenomenon and to interrogate the frequency of private exons originating from weakly spliced exons, independent of their possible association with disease. Consequently, we modelled the distribution of $\Psi$ for the 1,603,042 splicing events detected genome-wide in 105 samples as a mixture of three components. The model classified splicing frequencies per splice site as strong (20%, with $\Psi > 5.3\%$), weak (16%, with $0.16\% < \Psi < 5.3\%$), or background (64%, with $\Psi < 0.16\%$, Methods section, Fig. 3f and Supplementary Fig. 6). Strikingly, the majority (70%, 4.4-fold more than by chance) of the 17 discovered private exons originated from weak splice sites (Fig. 3f, bottom). These data confirm that weakly spliced cryptic exons are loci more susceptible to turn into strongly spliced sites than other intronic regions. These weak splicing events are usually dismissed as 'noise' since they are only supported by few reads in a given sample. Our analysis shows that they can be detected as accumulation points across multiple individuals. Hence, these results suggest that the prioritization of deep intronic VUSs gained through WGS could be improved by annotating weak splice sites and their resulting cryptic exons.

**Mono-allelic expression.** As a third approach, we searched for MAE of rare variants. In median per sample, 35,521 heterozygous SNVs were detected by WES, of which 7,622 were sufficiently covered by RNA-seq to call MAE (more than 10 reads), 20 showed MAE (Hochberg adjusted $P$ value < 0.05, allele frequency ≥ 0.8), of which 6 were rare variants (minor allele frequency < 0.001, Methods section, Fig. 4a, 90% of the samples had < 12 MAE of rare variants). Amongst the 18 rare mono-allelic expressed variants in patient #80256 was a VUS in *ALDH18A1* (c.1864C > T, p.Arg622Trp, Fig. 4b), encoding an enzyme involved in mitochondrial proline metabolism[45]. This VUS had been seen in WES compound heterozygous with a nonsense variant (c.1988C > A, p.Ser663*, Fig. 4b and Supplementary Fig. 7). Variants in *ALDH18A1* had been reported to be associated with cutis laxa III (OMIM #138250)[46,47], yet the patient did not present cutis laxa. Because of this inconsistent phenotype and the unknown significance of the non-synonymous variant, the variants in *ALDH18A1* were not regarded as disease causing. However, RNA-seq-based aberrant expression (Supplementary Fig. 7) and MAE analysis prioritized *ALDH18A1* again. Validation by quantitative proteomics revealed severe reduction down to ~2% ALDH18A1 (Fig. 4c), indicating that the rare MAE variant affects translation or protein stability. Metabolomics profile of blood plasma was in accordance with a defect in proline metabolism (Fig. 4d, Supplementary Method 8) and the following changes in urea cycle. Patient fibroblasts showed a growth defect that was rescued by supplementation of proline, validating impaired proline metabolism as the underlying molecular cause (Fig. 4e). Our experimental evidence strongly suggests that the two observed variants are pathogenic. Finally, a recent report[48] on *ALDH18A1* patients extended the phenotypic spectrum to spastic paraplegia without cutis laxa (OMIM #138250). Spastic paraplegia resembles the symptoms

of our patient (Supplementary Note 1), which validates these *ALDH18A1* mutations as disease-causing.

In another patient (#62346) we found borderline non-significant low expression of *MCOLN1* with 10 out of 11 reads expressing an intronic VUS (c.681-19A > C, Fig. 4f). This intronic variant was detected as part of a retained intron, which introduced a nonsense codon (p. Lys227_Leu228ins16*, Fig. 4f and Supplementary Fig. 8). When looking at the WES data we could additionally identify a heterozygous nonsense variant (c.832C > T, p.Gln278*). The allele with the exonic nonsense mutation was not expressed, most likely due to nonsense-mediated decay. Mutations in *MCOLN1* are associated with mucolipidosis (OMIM #605248). The symptoms of the patient were initially suggestive for mucolipidosis, but none of the enzymatic tests available for mucolipidosis types 1, 2 and 3 revealed an enzyme deficiency in blood leucocytes (Supplementary Note 1). Moreover, *MCOLN1* was missed by our WES variant filter since the intronic variant was not prioritized. Hence, the WES data could not be conclusive about *MCOLN1*. In contrast, the RNA-seq data demonstrated two loss-of-function alleles in *MCOLN1* and therefore established the genetic diagnosis.

**RNA defects in exome-diagnosed patients.** Here, we included genetically diagnosed patients in our RNA-seq analysis pipeline to increase the power for the detection of aberrant expression and aberrant splicing in fibroblast cell lines. However, when evaluating the results for 40 diagnosed cases with WES and RNA-seq available (Supplementary Table 1), aberrant splicing detected 7 out of 8 cases with a causal splicing variant, MAE recovered 3 out of 6 patients with heterozygous missense variants compound with a stop or frameshift variant, and aberrant expression recovered 3 out of 4 homozygous stop variants. Counter-intuitively, only one of the 9 frame-shift variants did lead to a detectable RNA defect, that is, MAE of a near splice site intronic variant within a retained intron. The partial recovery of stop and frameshift variants may reflect incomplete nonsense mediated decay. For none of the 14 genes where missense variants were disease causing, a RNA defect could be detected with our pipeline. This is expected, since missense variants more likely affect protein function rather than RNA expression (Supplementary Table 2).

**Discussion**

Altogether, our study demonstrates the utility of RNA-seq in combination with bioinformatics filtering criteria for molecular diagnosis by (i) discovering a new disease-associated gene, (ii) providing a diagnosis for 10% (5 of 48) of undiagnosed cases and (iii) identifying a limited number of strong candidates. We established a pipeline for the detection of aberrant expression, aberrant splicing and MAE of rare variants, that is able to detect significant outliers, with a median of 1, 5 and 6, respectively. Overall, for 36 patients our pipeline provides a strong candidate gene, that is, a known disease-causing or mitochondrial protein-encoding gene, like *MGST1* (Fig. 5a, Supplementary Data 7). This manageable amount, similar to the median number of 16 genes with rare potentially bi-allelic variants detected by WES, allows manual inspection and validation by disease experts. While filtering by frequency is highly efficient when focusing on the coding region, frequency filtering is not as effective for intronic or intergenic variants identified by WGS. The loss-of-function character observed on RNA level thus improved interpretation of VUS identified by genotyping.

We focused our analysis on one sample preparation pipeline, which has several advantages. Based on our experience, expression outliers can only reliably be detected after extensive

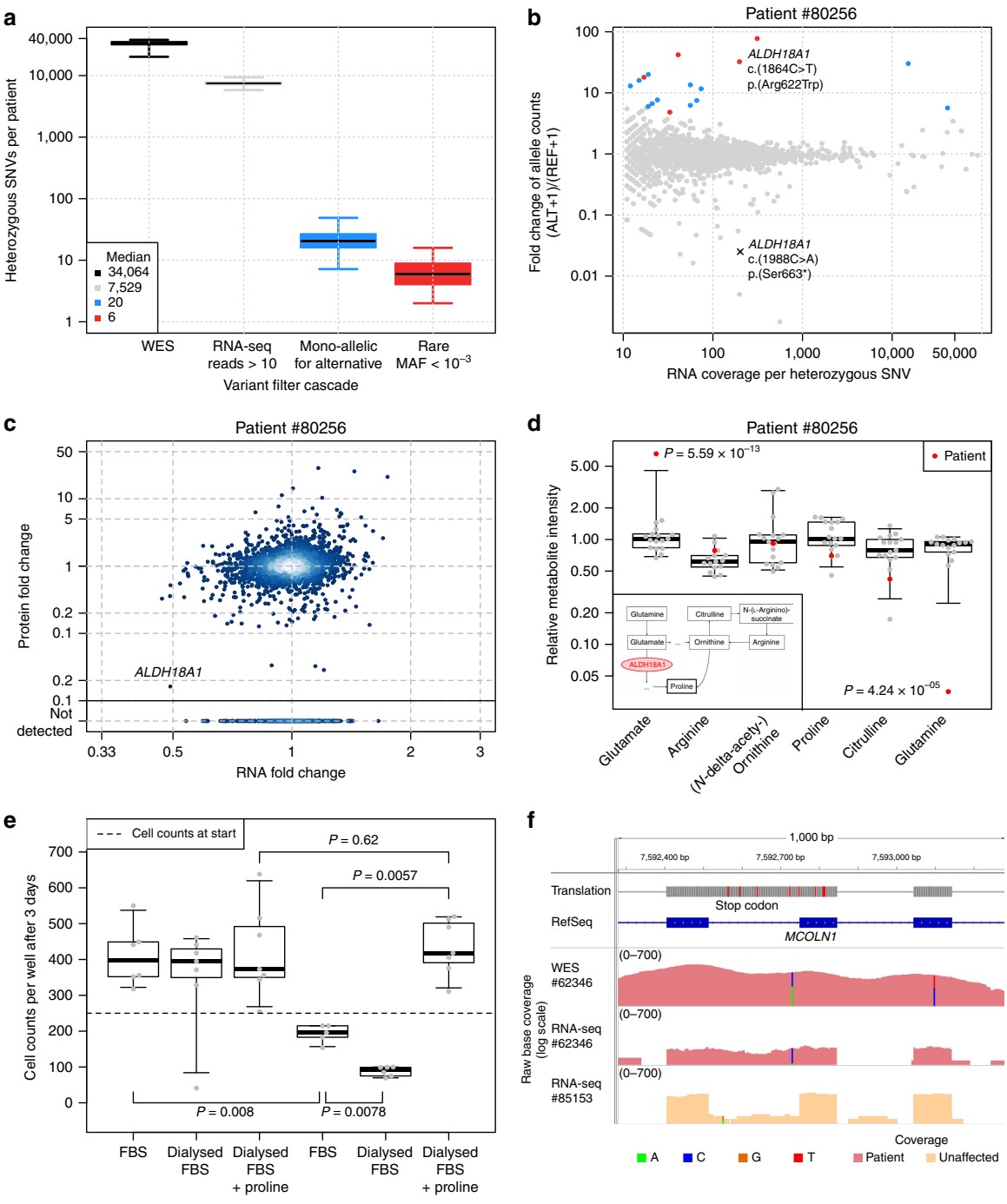

**Figure 4 | Detection and validation of MAE of rare variants.** (**a**) Distribution of heterozygous single nucleotide variants (SNVs) across samples for different consecutive filtering steps. Heterozygous SNVs detected by exome sequencing (black), SNVs with RNA-seq coverage of at least 10 reads (grey), SNVs where the alternative allele is mono-allelically expressed (alternative allele frequency > 0.8 and Benjamini-Hochberg corrected $P$ value < 0.05, blue), and the rare subset of those (ExAC minor allele frequency < 0.001, red). (**b**) Fold change between alternative (ALT + 1) and reference (REF + 1) allele read counts for the patient #80256 compared to total read counts per SNV within the sample. Points are coloured according to the groups defined in **a**. (**c**) Gene-wise comparison of RNA and protein fold changes of the patient #80256 compared to the average across the fibroblast cell lines of all other patients. The position of the gene *ALDH18A1* is highlighted. Reliably detected proteins that were not detected in this sample are shown separately with their corresponding RNA fold changes (points below solid horizontal line). (**d**) Relative intensity for metabolites of the proline biosynthesis pathway (inlet) for the patient #80256 and 16 healthy controls of matching age. Equi-tailed 95% interval (whiskers), 25th, 75th percentile (boxes) and median (bold horizontal line) are indicated. Data points belonging to the patient are highlighted (red circles, $P$ values were computed using the Student's $t$-test). (**e**) Cell counts under different growth conditions for the NHDF and patient #80256. Both fibroblasts were grown in fetal bovine serum (FBS), dialysed FBS (without proline) and dialysed FBS with proline added. Boxplot as in **d**. $P$ values are based on a two-sided Wilcoxon test. (**f**) Intron retention for *MCOLN1* in patient #62346. Tracks from top to bottom: genomic position on chromosome 19, amino acid translation (red for stop codons), RefSeq gene model, coverage of WES of patient #62346, RNA-seq based coverage for patients #62346 and #85153 (red and orange shading, respectively). SNVs are indicated by non-reference coloured bars with respect to the corresponding reference and alternative nucleotide.

normalization process. This needs information about all technical details starting from the biopsy, growth of the cells, to the RNA extraction and library preparation. Usually not all this information is available in published data sets. For detecting aberrant splicing such as new exons, we would recommend not to mix different tissues because splicing can be tissue-specific. MAE is the most robust of all criteria in this respect because it only relies on read counts within a sample. Overall, we recommend not relying on a single sample being compared to public RNA-seq data sets. Instead, RNA-seq should be included in the pipeline of diagnostic centres to generate matching controls over time. The situation is similar for whole exome and WGS, where the control for platform-specific biases is important.

To our surprise, many newly diagnosed cases were caused by a defective splicing event, which caused loss of function (Fig. 5b), confirming the increasing recognition of the role of splicing defects in both Mendelian[49,50] and common disorders[25]. In the case of *TIMMDC1*, the causal variant was intronic, and thus not covered by WES. Even when detected by WGS, such deep intronic variants are difficult to prioritize from the sequence information alone. Here, we showed that RNA-seq of large cohorts can provide important information about intronic positions that are particularly susceptible to affect splicing when mutated. We showed that private exons often arise from loci with weak splicing of ∼1%. This suggests that rare variants affecting such cryptic splice sites are more likely to affect splicing and that these can be detected as positions with low yet consistent splicing. We reason that analysis of a RNA-seq compendium of healthy donors across multiple tissues such as GTEx (ref. 51) could provide tissue-specific maps of cryptic splice sites useful for prioritizing intronic variants.

Genetic disorders typically show specificity to some tissues, some of which might not be easily accessible for RNA-sequencing. It is therefore natural to question whether transcriptome sequencing of an unaffected tissue can help diagnosis. Here, we performed RNA-seq on patient derived dermal fibroblast cell lines. The fibroblast cell lines are the byproducts of muscle biopsies routinely undertaken in the clinic to biochemically diagnose mitochondrial disorders with enzymatic assays. By using fibroblast cell lines we overcome the limited accessibility of the affected tissues, which in the case of mitochondrial disorders are often high energy demanding tissues like brain, heart, skeletal muscle or liver. It turns out that many genes with a mitochondrial function are expressed in most tissues[52], including fibroblasts. Hence, extreme regulatory defects such as loss of expression or aberrant splicing of genes encoding mitochondrial proteins can be detected in fibroblasts, even though the physiological consequence on fibroblasts might be negligible. This property might be true for other diseases: the tissue-specific physiological consequence of a variant does not necessarily stem from tissue-specific expression of the gene harbouring the variant. In many cases, tissue-specificity might be due to environmental or cellular context, or from tissue-specific expression of further genes. Hence, tissue-specificity does not preclude RNA-seq of unaffected tissues from revealing the causative defect for a large number of patients. Moreover, non-affected tissues have the advantage that the regulatory consequences on other genes are limited and therefore the causal defects are more likely to stand out as outliers[53].

Parallel to our effort, another study systematically investigated the usage of RNA-seq for molecular diagnosis with a similar sample size, using muscle biopsies from primary muscle disorder patients[50]. Analogously to our approach, not only exome sequencing-based VUS candidates were validated, but also new disease-causing mechanisms identified using RNA-seq data.

Despite a few differences in the approach (expression outliers were not looked for, only samples of the affected tissues were considered and using samples of healthy donors as controls), the results are in line with ours whereby aberrant splicing also turns out to be a frequent disease-causing event. The success rate was even higher (35%), possibly because the diagnostic rate of primary muscle disorders is higher than for mitochondrial disorders. Also, in this case the affected tissue was always accessible and could be profiled. Therefore the chances were higher that the affected gene is expressed. Altogether, this complementary study confirms the relevance of using RNA-seq for diagnosis of Mendelian disorders.

In conclusion, we predict that RNA-seq will become an essential companion of genome sequencing to address undiagnosed cases of genetic disease.

## Methods

**Study cohort.** All individuals or their guardians gave written informed consent before undergoing evaluation and testing, in agreement with the Declaration of Helsinki and approved by the ethical committees of the centres participating in this study, where biological samples were obtained. All studies were completed according to local approval of the ethical committee of the Technical University of Munich. The study cohort is described in detail in Supplementary Note 1.

**Exome sequencing.** DNA from fibroblast cell lines was isolated from whole-cell lysates using the AllPrep DNA Mini Kit (Qiagen, Hilden, Germany) according to the manufacturer's protocol. Exonic regions were enriched using the SureSelect Human All Exon kit from Agilent (Supplementary Data 3) followed by sequencing as 100 bp paired-end runs on an Illumina HiSeq2000 and Illumina HiSeq2500 (AG_50MB_v4 and AG_50MB_v5 exome kit samples) or as 76 bp paired-end runs on the Illumina GAIIx (AG_38MB_v1 and AG_50MB_v3 exome kit samples)[54].

**Exome alignment and variant prioritization.** Read alignment to the human reference genome (UCSC Genome Browser build hg19) was done using Burrows-Wheeler Aligner[55] (v.0.7.5a). Single-nucleotide variants and small insertions and deletions (indels) were detected with SAMtools[56,57](version 0.1.19). Variants with a quality score below 90, a genotype quality below 90, a mapping quality below 30, and a read coverage below 10 were discarded. The reported variants and small indels were annotated with the most severe entry by the Variant Effector Predictor[58] based on The Sequence Ontology term ranking[59]. The candidate variants for one patient are filtered to be rare, affect the protein sequence and potentially both alleles.

Variants are rare with a minor allele frequency <0.001 within the ExAC database[33] and a frequency <0.05 among our samples. Our filter considers variants to affect the protein, if they are a coding structural variant or their mutation type is one of ablation, deletion, frame-shift, incomplete, start lost, insertion, missense, splice, stop gain, stop retain, unstart and unstop. A potential biallelic effect can be caused by either a homozygous or at least two heterozygous variants in the same gene, whereas in latter case we assume that the heterozygous variants are on different alleles (Supplementary Fig. 9). This filter is designed for a recessive type disease model and does not account for a single heterozygous variant that could be disease causing in a dominant way.

**Variant of unknown significance.** 'A variation in a genetic sequence whose association with disease risk is unknown. Also called unclassified variant, variant of uncertain significance and VUS.' (see https://www.cancer.gov/publications/dictionaries/genetics-dictionary?cdrid=556493).

**Cell culture.** Primary patient fibroblast cell lines, normal human dermal fibroblasts (NHDF) from neonatal tissue (Lonza) and 293FT cells (Thermo Fisher Scientific) were cultured in high glucose DMEM (Life Technologies) supplemented with 10% FBS, 1% penicillin/streptomycin, and 200 µM uridine at 37 °C and 5% $CO_2$. All fibroblast cell lines have been tested negative for mycoplasma contamination.

**RNA sequencing.** Non-strand specific, polyA-enriched RNA-seq was performed as described in ref. 23: RNA was isolated from whole-cell lysates using the AllPrep RNA Kit (Qiagen) and RNA integrity number (RIN) was determined with the Agilent 2100 BioAnalyzer (RNA 6000 Nano Kit, Agilent). For library preparation, 1 µg of RNA was poly(A) selected, fragmented and reverse transcribed with the Elute, Prime and Fragment Mix (Illumina). End repair, A-tailing, adaptor ligation and library enrichment were performed as described in the Low Throughput protocol of the TruSeq RNA Sample Prep Guide (Illumina). RNA libraries were

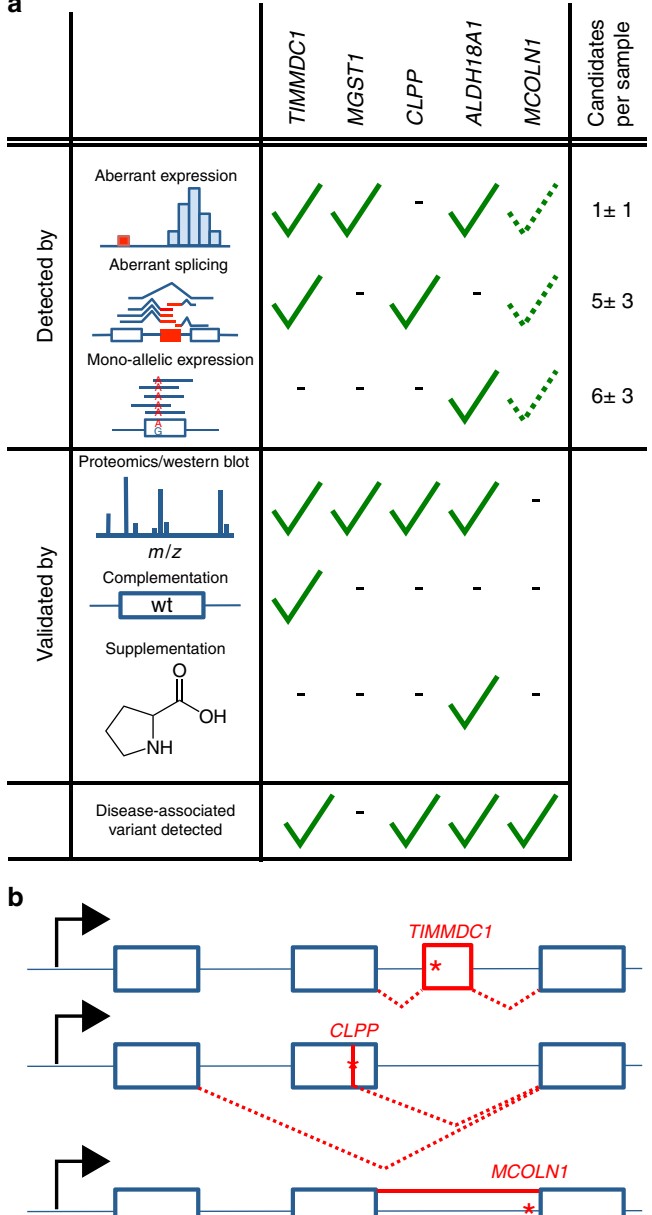

**Figure 5 | Characterization of diagnoses and variants causing aberrant splicing.** (**a**) Detection strategy and validation of genes with RNA defects in newly diagnosed patients, that is, *TIMMDC1* ($n = 2$ patients), *CLPP*, *ALDH18A1* and *MCOLN1*, and one patient with a strong candidate, that is, *MGST1*. The median number ($\pm$ median absolute deviation) of candidate genes is given per detection strategies. Dotted check: identified by manual inspection (not statistically significant). (**b**) Schematic representation of variant causing splicing defects for *TIMMDC1* (top, new exon red box), *CLPP* (middle, exon skipping and truncation) and *MCOLN1* (bottom, intron retention). Variants are depicted by a red star.

assessed for quality and quantity with the Agilent 2100 BioAnalyzer and the Quant-iT PicoGreen dsDNA Assay Kit (Life Technologies). RNA libraries were sequenced as 100 bp paired-end runs on an Illumina HiSeq2500 platform.

**Processing of RNA sequencing files.** RNA-seq reads were demultiplexed and mapped with STAR[60] (version 2.4.2a) to the hg19 genome assembly (UCSC Genome Browser build). In addition to the default parameters we detected gene fusions and increased sensitivity for novel splice junctions (chimSegmentMin = 20, twopassMode = 'Basic'). Analysis was restricted to the 27,682 UCSC Known Genes[61] (genome annotation version hg19) of chromosomes 1–22, M, X or Y. Per

gene, reads that are paired with mates from opposite strands and that overlapped completely within the gene on either strand orientation were counted using the summarizeOverlaps function of the R/Bioconductor GenomicAlignments[62] package (parameters: mode = intersectionStrict, singleEnd = FALSE, ignore.strand = TRUE and fragments = FALSE). If the 95th percentile of the coverage across all samples was below 10 reads the gene was considered 'not expressed' and discarded from later analysis.

**Computing RNA fold changes and differential expression.** Before testing for differential expression between one patient of interest and all others, we controlled for technical batch effect, sex and biopsy site as inferred from the expression of *hox* genes (Supplementary Method 1, Supplementary Data 8). We modelled the RNA-seq read counts $K_{i,j}$ of gene $i$ in sample $j$ with a generalized linear model using the R/Bioconductor DESeq2 package[63,64]:

$$K_{i,j} \sim \mathrm{NB}\left(s_j \times q_{i,j}, \alpha_i\right)$$

$$\log_2(q_{i,j}) = \beta_i^0 + \beta_i^{\mathrm{condition}} \mathbf{x}_{i,j}^{\mathrm{condition}} + \beta_i^{\mathrm{batch}} \mathbf{x}_{i,j}^{\mathrm{batch}} + \beta_i^{\mathrm{sex}} \mathbf{x}_{i,j}^{\mathrm{sex}} + \beta_i^{\mathrm{hox}} \mathbf{x}_{i,j}^{\mathrm{hox}}$$

Where NB is the negative binomial distribution; $\alpha_i$ is a gene specific dispersion parameter; $s_j$ is the size factor of sample $j$; $\beta_i^0$ is the intercept parameter for gene $i$. The value of $\mathbf{x}_{i,j}^{\mathrm{condition}}$ is 1 for all RNA samples $j$ of the patient of interest, thereby allowing for biological replicates, and 0 otherwise. The resulting vector $\beta_i^{\mathrm{condition}}$ represents the $\log_2$ fold changes for one patient against all others. Z-scores were computed by dividing the fold changes by the s.d. of the normalized expression level of the respective gene. The $P$ values corresponding to the $\beta_i^{\mathrm{condition}}$ were corrected for multiple testing using the Hochberg family-wise error rate method[65].

**Detection of aberrant splicing.** The LeafCutter[66] software was utilized to detect aberrant splicing. Each patient was tested against all others. To adjust LeafCutter to the rare disease setting, we modified the parameters to detect rare clusters, capture local gene fusion events and to detect junctions unique to a patient (minclureads = 30; maxintronlen = 500,000; mincluratio = 1e-5, Supplementary Data 9). Furthermore, one sample was tested against all other samples (min_samples_per_group = 1; min_samples_per_intron = 1). The resulting $P$ values were corrected for multiple testing using a family-wise error rate approach[65].

The significant splice events (Hochberg adjusted $P$ value < 0.05) detected in the undiagnosed patients were visually classified as exon skipping, exon truncation, exon elongation, new exon, complex splicing (any other splicing event or a combination of the aforementioned ones) and false positives ($n = 73$, Fig. 3b). However, due to LeafCutter's restriction to split reads it is difficult to detect intron retention events, since no split-read is present in a perfect intron-retention scenario.

For further analysis, only reads spanning a splice junction, so called split reads, were extracted with a mapping quality of > 10 to reduce the false-positive rate due to mapping issues. Each splice site was annotated as belonging to the start or end of a known exon or to be entirely new. For the reference exon annotation the GENCODE release 24 based on GRCh37 was used[67]. The per cent spliced in ($\Psi$) values[32] for the 3' and 5' sites were calculated as:

$$\Psi_5(D, A) = \frac{n(D, A)}{\sum_{A'} n(D, A')} \quad \text{and} \quad \Psi_3(D, A) = \frac{n(D, A)}{\sum_{D'} n(D', A)}$$

where $D$ is a donor site and $A$ is an acceptor site. $n(D,A)$ denotes the number of reads spanning the given junction. $D'$ and $A'$ represent all possible donor and acceptor sites, respectively.

Classification of splice sites into background, weak and strong was done by modelling the distribution of the $\Psi_5$ and $\Psi_3$-values with three components. Identifiability of the three components was facilitated by considering three groups of junctions depending on previous annotation of splice sites: 'no side is annotated', 'one side is annotated' and 'both sides are annotated'. Specifically, the number of split reads $n(D, A)$ of a junction conditioned on the total number of reads $N(D, A) = \Sigma_{A'} n (D, A')$, for $\Psi_5$, and $N (D, A) = \Sigma_{D'} n (D', A)$, for $\Psi_3$, was modelled as:

$$P(n(D, A) \mid N(D, A)) = \sum_{c \in \{bg, wk, st\}} \sum_{s=0,1,2} \pi_{s,c} \mathrm{BetaBin}(n(D, A) \mid N(D, A), \alpha_c, \beta_c)$$

where $c$ is the component index, $s$ the number of annotated sites and BetaBin the beta-binomial distribution. Hence, the components were modelled to have the same parameters $\alpha_c, \beta_c$ in all three groups but their mixing proportions $\pi_{s,c}$ to be group-specific. Fitting was performed using the expectation-maximization algorithm. For the initial step, the data points were classified as background ($\Psi < 0.001$), weak spliced ($\Psi < 0.1$) and canonical ($\Psi > = 0.1$). After convergence of the clustering the obtained parameters were used to estimate the probability for each junction side to belong to a given class.

**Detection of mono-allelic expression.** For MAE analysis only heterozygous single-nucleotide variants with only one alternative allele detected from exome sequencing data were used. The same quality filters were used as mentioned in the exome sequencing part, but no frequency filter was applied. To get allele counts from RNA-seq for the remaining variants the function pileLettersAt from the R/Bioconductor package GenomicAlignments[62] was used. The data were further filtered for variants with coverage of at least 10 reads on the transcriptome.

The DESeq2 package[63,64] was applied on the final variant set to estimate the significance of the allele-specific expression. Allele-specific expression was estimated on each heterozygous variant independently of others (that is, without phasing the variants). For each sample, a generalized linear model was fitted with the contrast of the coverage of one allele against the coverage of the other alleles. Specifically, we modelled $K_{i,j}$ the number of reads of variant $i$ in sample $j$ as:

$$K_{i,j} \sim NB(s_j \times q_{i,j}, \alpha)$$

$$\log_2(q_{i,j}) = \beta_i^0 + \beta_i^{allele} \mathbf{x}_{i,j}^{allele}$$

Where NB is the negative binomial distribution; the dispersion parameter $\alpha$ was fixed for all variants to $\alpha = 0.05$, which is approximately the average dispersion over all samples based on the gene-wise analysis; $s_j$ is the size factor of each condition; $\beta_i^0$ is the intercept parameter for variant $i$. The value of $\mathbf{x}_{i,j}^{allele}$ is 1 for the alternative alleles and 0 otherwise. The resulting $\beta_i^{allele}$ represents the $\log_2$-fold changes for the alternative allele against the reference allele. The independent filtering by DESeq2 was disabled (independentFiltering = FALSE) to keep the coverage outliers among the results. To classify a variant as mono-allelically expressed a cutoff of $|\beta_i^{condition}| \geq 2$ was used, which corresponds to an allele frequency $\geq 0.8$, and we filtered Benjamini-Hochberg adjusted $P$ values to be $< 0.05$.

**Transduction and transfection.** Overexpression of *TIMMDC1* in fibroblast cell lines was performed by lentivirus-mediated expression of the full-length *TIMMDC1* cDNA (DNASU Plasmid Repository) using the ViraPower HiPerform Lentiviral TOPO Expression Kit (Thermo Fisher Scientific)[68]. *TIMMDC1* cDNA was cloned into the pLenti6.3/V5-TOPO expression vector and cotransfected into 293FT cells with the packaging plasmid mix using Lipofectamine 2000. After 24 h, the transfection mix was replaced with high glucose DMEM supplemented with 10% FBS. After further 72 h, the viral particle containing supernatant was collected and used for transduction of the fibroblast cell lines. Selection of stably expressing cells was performed using 5 µg ml$^{-1}$ Blasticidin (Thermo Fisher Scientific) for 2 weeks.

**Immunoblotting.** Total fibroblast cell lysates were subjected to whole protein quantification, separated on 4–12% precast gels (Lonza) by SDS–polyacrylamide gel electrophoresis (PAGE) electrophoresis and semi-dry transferred to polyvinylidene difluoride membranes (GE Healthcare Life Sciences). The membranes were blocked in 5% non-fat milk (Bio Rad) in TBS-T (150 mM NaCl, 30 mM Tris base, pH 7.4, 0.1% Tween 20) for 1 h and immunoblotted using primary antibodies (1:1,000 dilution) against CLPP (Abcam, ab56455), MCOLN1 (Abcam, ab28508), MT-ND5 (Abcam, ab92624), NDUFA13 (Abcam, ab110240), NDUFB3 (Abcam, ab55526), NDUFB8 (Abcam, ab110242), TIMMDC1 (Abcam, ab171978) and UQCRC2 (Abcam, ab14745) for 1 h at RT or ON at 4 °C. Signals were detected by incubation with horseradish peroxidase (HRP)-conjugated goat anti-rabbit and goat anti-mouse secondary antibodies (Jackson Immuno Research Laboratories, Code: 111-036-045 and Code: 115-036-062, respectively, 1:5,000 dilution) for 1 h and visualized using ECL (GE Healthcare Life Sciences).

**Blue native PAGE (BN-PAGE).** Fresh fibroblast cell pellets were resuspended in PBS supplemented with 0.25 mM PMSF and 10 U ml$^{-1}$ DNASe I and solubilized using 2 mg digitonin per mg protein. The mixture was incubated on ice for 15 min followed by addition of 1 ml PBS and subsequent centrifugation for 10 min at 10,000*g* and 4 °C. The pellet was resuspended in 1x MB (750 mM ε-aminocaproic acid, 50 mM bis-Tris, 0.5 mM EDTA, pH 7.0) and subjected to whole-protein quantification. Membrane proteins were solubilized at a protein concentration of 2 µg µl$^{-1}$ using 0.5% (v/v) *n*-dodecyl-β-d-maltoside for 1 h on ice and centrifuge for 30 min at 10,000*g* at 4 °C. The supernatant was recovered and whole protein amount was quantified. Serva Blue G (SBG) was added to a final concentration of 0.25% (v/v) and 60 µg protein were loaded on NativePAGE 4–16% Bis-Tris gels (Thermo Fisher Scientific). Anode buffer contained 50 mM Bis-Tris, pH 7.0, blue cathode buffer contained 15 mM Bis-Tris, 50 mM Tricine, pH 7.0, 0.02% SBG. Electrophoresis was started at 40 V for 30 min and continued at 130 V until the front line proceeded 2/3 of the gel. Subsequently, blue cathode buffer was replaced by clear cathode buffer not containing SBG (15 mM Bis-Tris, 50 mM Tricine, pH 7.0). Proteins were wet transferred to polyvinylidene difluoride membranes and immunoblotted using primary antibodies against NDUFB8 (Abcam, ab110242, 1:1,000 dilution) to visualize complex I and UQCRC2 (Abcam, ab14745, 1:1,000 dilution) to visualize complex III. Incubation with secondary antibodies and detection was performed as described for immunoblotting.

**Proline supplementation growth assay.** We modified a method established earlier[46]. For the comparative growth assay, equal number of cells ($n = 250$) from patient and control were seeded in 96-well plates and cultured in DMEM containing 10% of either normal or dialysed FBS. Medium with normal FBS contains small molecules, whereas medium with dialysed FBS is free of molecules with a molecular weight smaller than 10,000 mw (Proline-free medium). To confirm the effect of Proline deprivation, DMEM containing dialysed FBS was supplemented with 100 µM ʟ-Proline to rescue the growth defect. After paraformaldehyde fixation, nuclei were stained with 4′,6-diamidino-2-phenylindole and cell number was determined using a Cytation3 automated plate reader (BioTek, USA).

**Data availability.** All data and R scripts needed to reproduce paper figures are available online at our webserver (https://i12g-gagneurweb.in.tum.de/public/paper/mitoMultiOmics). Additional data are available on request due to privacy or other restrictions.

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

## Acknowledgements

We are grateful to the participating families. Further we thank Dr Lina Florentin for providing DNA samples of the parents of #35791. This study was supported by the German Bundesministerium für Bildung und Forschung (BMBF) through the E-Rare project GENOMIT (01GM1603 and 01GM1207, H.P. and T.M., FWF I 920-B13 for J.A.M. and J41J11000420001 for D.G.), through the Juniorverbund in der Systemmedizin 'mitOmics' (FKZ 01ZX1405A, D.M.B. and J.G., and FKZ 01ZX1405C, T.B.H.), and the DZHK (German Centre for Cardiovascular Research, L.S.K., T.M.). The study was furthermore supported by the Deutsche Forschungsgemeinschaft (German Research Foundation) within the framework of the Munich Cluster for Systems Neurology (EXC 1010 SyNergyDFG, T.M.) and a Fellowship through the Graduate School of Quantitative Biosciences Munich (QBM) supports D.M.B.. H.P., V.T. and J.A.M. are supported by EU FP7 Mitochondrial European Educational Training Project (317433). C.M., R.K., J.G. and H.P. are supported by EU Horizon2020 Collaborative Research Project SOUND (633974). R.W.T. is supported by the Wellcome Centre for Mitochondrial Research (203105/Z/16/Z) and the MRC Centre for Neuromuscular Diseases (G0601943). D.G. is supported by Telethon-Italy (GGP15041). We thank the Pierfranco and Luisa Mariani Foundation and the Cell lines and DNA Bank of Paediatric Movement Disorders and Mitochondrial Diseases of the Telethon Genetic Biobank Network (GTB09003).

## Author contributions

T.M., J.G. and H.P. planned the project. J.G. and H.P. overviewed the research. H.P. designed the experiments. C.L., B.F., A.D., V.T., A.L., D.G., R.W.T., D.G., J.A.M., A.R., P.F., F.D. and T.M. reviewed the phenotypes, performed sample collection and biochemical analysis. L.S.K., D.M.B., C.M., T.M.S. and H.P. curated and analysed the data. J.G. devised the statistical analysis. L.S.K., R.K., A.I., C.T., E.K. and B.R. performed the cell biology experiments. L.S.K., R.K., E.G., T.S., P.L. and T.M.S. performed exome, genome and RNA-seq. L.S.K., R.K., T.B.H. and H.P. performed the exome analysis. L.S.K. and G.P. performed the quantitative proteomics experiments. L.S.K., G.K. and J.A. performed the metabolomics studies. L.S.K., D.M.B., C.M., J.G. and H.P. wrote the manuscript. L.S.K., D.M.B. and C.M. visualized the data. Critical revision of the manuscript was performed by all authors.

## Additional information

**Competing interests:** The authors declare no competing financial interests.

