## [Peer Review File · Nature Communications]

Reviewers' comments:

Reviewer #1 (Remarks to the Author):

None

Reviewer #3 (Remarks to the Author):

With great interest I again read the (revised) version of the manuscript by Kremer & Bader et al. entitles "Genetic diagnosis of Mendelian disorders via RNA sequencing". In this version, the authors have addressed many of my questions coming forward from my original review of the manuscript, but point unfortunately still remain unclear.

- The authors provided many additional tables which I very much appreciated. Despite all this additional information I however still cannot find an overview of all 105 patient Ids linked to the information whether they had a diagnosis (or not, and if so, what diagnosis did they have?), whether they had WED (or not) whether RNA sequencing provided the diagnosis (or not), whether the WES diagnosis was confirmed in the RNA sequencing information (or not), whether proteomics was performed (or not) etc etc

Table S1 contains a summary of this information, but without the actual patient Ids, this cannot be reproduced or interpreted to its full extent.

- Table S2 contains information on 48 samples. Why only data on these 48 and not all 105? Does this mean that for the remaining 57 samples, no aberrant expression, splicing or MAE was identified?

- The authors provide the medians for genes with aberrant expression (n=1), aberrant splicing (n=5) and MAE (n=6), which I assume, originate from Table S2. This can however not easily be derived from this table. The authors must provide another supplementary table listing these details for all 105 patients examined. In detail, this should contain the sample ID, with number of genes with aberrant expression, splicing and MAE.

- To verify the authors observations on the median number of splice defects (for the 48 samples provided in Suppl Table 2) I noted that the median number of splice defects per patient is not 5 but 4 per patient. Notably, when preparing this table, I noted that there are 8 of 48 samples that show (extreme) outliers for the analysis performed: 68547, 80255 (for expression); 68547, 80248, 68558 and 81247 (for splicing) and 54485, 56711, 33281 and 68540 for MAE. Can the authors explain these extreme outliers? Also, when presenting the median of aberrant products in the main text, provide the ranges for the observations (e.g. for expression: median of 1 gene per sample; range 0-493).

- In the results section, the authors mention that for 105 samples RNA sequencing has been performed, but the data analysis seems to be restricted to the 48 samples without a diagnosis. Yet, in the discussion, the authors mention the analyses for the samples where WES led to a diagnosis. They report that for missense mutations, the RNA-seq does not help the analysis as only limited effect on expression level, splicing and/or MAE is observed. Please add these analyses to the results section, as these would be a primary/starting point if one would start evaluating RNAseq (as potential positive controls). In the discussion, one can then still mention that there is no added benefit as most mutations did not lead to the effects analysed, but perhaps one can add the analysis of 'routine variant detection' as performed in WES analysis? If all (diagnostically relevant) WES mutations can be identified using 'simple variant calling on RNA data, one could argue that RNAseq is more efficient

than WES for some diseases?

- Based on the supplementary tables, it is my understanding that there are 47 sample Ids for which there is a significant WES variant (ranging from 5 to 293 variants per sample), but where there is no effect in the RNA analysis performed. Can the authors comment on this? Also, please explain how one sample has 293 significant WES variants? The latter sample, 33281, was also one of the outliers in the analysis for MAE....It suggests that this sample does not meet the quality specs, or if so, that the metrics used may leave sample(s) that should be removed?

- The authors comment on the paper by Cummings et al. in their discussion. They correctly report that there is a discrepancy in '% patients solved' in both studies (10% here compared to 35% by Cummings et al.). Please discuss why these are so different! The authors do comment that the fact that both studies successfully identify diagnosis shows the value of RNAseq, but this obvious conclusion must be complemented by discussion on the rationale for the differences in percentage diagnosis.

- Figure 1 contains a circle with two shades of blue for 'diagnosis' and 'no diagnosis' on the top, and in addition to those colours, a shaded version on the bottom. The shaded part is likely to represent the 10% diagnoses? But cartoon shows shading that is more than 10%. I also believe that perhaps these pie charts are better to be included in the last figure, where the summarize the results.

- I cannot help to be left with the feeling that some of the findings presented (such as the ALDH18A1 and MCOLN1) WES already pointed towards the molecular diagnosis, and that RNAseq was confirming this diagnosis, rather than identifying it. That is, if the molecular diagnostic lab performing WES would have ask for routine RNA work-up of these genes, the diagnosis would have also been performed, without the need of 'full RNAseq'.

REVIEWERS' COMMENTS:

Reviewer #3 (Remarks to the Author):

I wish to start with an apology for not meeting the deadline for completing this review. But mostly, I would like to thank the authors for providing the detailed responses to my queries. They have been addressed appropriately and I have not further comments for review.

Point-by-point response to the reviewers

Manuscript: "Genetic diagnosis of Mendelian disorders using RNA sequencing"
Reviews on NCOMMS-16-30508-T from March 23rd 2017.

Reviewer 1:

Referee 1 also noted that while many of his/her earlier technical concerns were addressed fully, some issues were not. For example, s/he felt that your response to the issue of ascribing pathogenicity when the phenotype is discordant was not satisfying. We would like you to consider this point when revising paper, especially given referee 3's request for clarification on patient information/diagnosis status, and incorporating RNAseq data you already have on missense mutations.

Answer:

With a recently published paper, our ALDH18A1 patient fits to the described genotype-phenotype correlation. We had included this reference in the former revised version but realized that the text was maybe not clear enough. We have extended our explanations of ALDH18A1 apparent phenotypic discordance. We address the questions of referee 3 below.

Response to Reviewer 3

With great interest I again read the (revised) version of the manuscript by Kremer & Bader et al. entitled "Genetic diagnosis of Mendelian disorders via RNA sequencing". In this version, the authors have addressed many of my questions coming forward from my original review of the manuscript, but point unfortunately still remain unclear.

Q1

The authors provided many additional tables which I very much appreciated. Despite all this additional information I however still cannot find an overview of all 105 patient IDs linked to the information whether they had a diagnosis (or not, and if so, what diagnosis did they have?), whether they had WED (or not) whether RNA sequencing provided the diagnosis (or not), whether the WES diagnosis was confirmed in the RNA sequencing information (or not), whether proteomics was performed (or not) etc etc. Table S1 contains a summary of this information, but without the actual patient IDs, this cannot be reproduced or interpreted to its full extent.

A: The reviewer is correct this information was not present in one table. Table S1 provided only candidate genes for unsolved patients. We now updated the sample annotation in Supplementary Table 1 to include all columns suggested by the reviewer. We also updated the corresponding description of all supplementary data and tables in our Supplementary information document.

Q2

Table S2 contains information on 48 samples. Why only data on these 48 and not all 105? Does this mean that for the remaining 57 samples, no aberrant expression, splicing or MAE was identified?

A: This table showed only information for the patients that were undiagnosed (n=48) before our study. We identified RNA defects of all kinds in the already diagnosed patients as now described in the last section of the new results. As stated in Q1, we now provide more information for all 105 patients in the sample annotation table Supplementary Table 1.

Q3

The authors provide the medians for genes with aberrant expression (n=1), aberrant splicing (n=5) and MAE (n=6), which I assume, originate from Table S2. This can however not easily be derived from this table. The authors must provide another supplementary table listing these details for all 105 patients examined. In detail, this should contain the sample ID, with number of genes with aberrant expression, splicing and MAE.

A: See Q1 for corresponding updates in Supplementary Table 1.

Q4

To verify the authors observations on the median number of splice defects (for the 48 samples provided in Suppl Table 2) I noted that the median number of splice defects per patient is not 5 but 4 per patient.

A: The reviewer is correct. For the 48 unsolved patients the median aberrant splicing defect is 4. Considering all 105 patients the median is 5. See Q1 for corresponding updates in Supplementary Table 1.

Q5

Notably, when preparing this table, I noted that there are 8 of 48 samples that show (extreme) outliers for the analysis performed: 68547, 80255 (for expression); 68547, 80248, 68558 and 81247 (for splicing) and 54485, 56711, 33281 and 68540 for MAE. Can the authors explain these extreme outliers?

A: The updated sample annotation table (Supplementary Table 1) provides directly the number of RNA defects per sample for the three performed analyses. Compared to the previous supplementary information, the counts are now given for all 105 samples. Indeed, sample #68547 shows a very large number of genes with aberrant expression (493). We could not find a technical or a biological explanation. It is an unsolved case, cells grow normally and the sample was sequenced at similar depth as the others. The high number of genes with aberrant expression observed in this patient may be related to the disorder. The maximum number of RNA defects stays reasonable for the other samples mentioned by the reviewer (18 for MAE for sample 68540 and 41 for splicing for sample 68547). In general, across all 105 samples, no sample shows very large numbers of MAE (maximum of 23), maybe because the two alleles are internal controls of each other. For the other two criteria, the figures 2a and 3a show the entire distribution. Both distributions have a long tail with 4 samples with more than 100 outliers (only sample 35834 is common to both). We could not trace these behaviors to a biological or a technical explanation at this stage.

Q6

Also, when presenting the median of aberrant products in the main text, provide the ranges for the observations (e.g. for expression: median of 1 gene per sample; range 0-493).

A: The figures (2a, 3a, and 4a) show the full range of the data. As this referee pointed out, some of our samples have aberrantly large numbers of such outliers and are therefore not representative. Consequently, we chose to report the 90th percentile rather than the full range in the figure, and now also in the text.

Q7

In the results section, the authors mention that for 105 samples RNA sequencing has been performed, but the data analysis seems to be restricted to the 48 samples without a diagnosis.

A: All analyses and plots have been performed on the full dataset (105 samples). Interpretation and follow ups have been performed on the unsolved cases only. The 57 diagnosed patients are now discussed in the results (see Q8). See Q1 for corresponding updates in Supplementary Table 1.

Q8

Yet, in the discussion, the authors mention the analyses for the samples where WES led to a diagnosis. They report that for missense mutations, the RNA-seq does not help the analysis as only limited effect on expression level, splicing and/or MAE is observed. Please add these analyses to the results section, as these would be a primary/starting point if one would start evaluating RNAseq (as potential positive controls).

A: We moved this part to the results section. We placed it as the last paragraph because it relies on all steps described before.

Q9

In the discussion, one can then still mention that there is no added benefit as most mutations did not lead to the effects analysed, but perhaps one can add the analysis of 'routine variant detection' as performed in WES analysis? If all (diagnostically relevant) WES mutations can be identified using 'simple variant calling on RNA data, one could argue that RNAseq is more efficient than WES for some diseases?

A: We would not suggest to perform RNA-seq for variant calling because RNA-seq is not comprehensive as not all genes are expressed in a given cell type or tissue and the affected tissue is only rarely available. We added it to the discussion when comparing to Cummings et al, bioRxiv.

Q10

Based on the supplementary tables, it is my understanding that there are 47 sample IDs for which there is a significant WES variant (ranging from 5 to 293 variants per sample), but where there is no effect in the RNA analysis performed. Can the authors comment on this?

A:

We also end up with 47 sample IDs (of the 48 unsolved patients) when searching for samples with at least one significant WES variant with no corresponding RNA effect. Some WES variants are missense and would likely not lead to a RNA effect. Some genes are not expressed in fibroblast cells (only about 12,000 are). This point is now stressed in the discussion when comparing to Cummings et al. Finally, we do not expect 100% sensitivity for our criteria. Because there are between 5 and 293 genes with significant WES variants in those samples, it

is not surprising that typically at least one is not recovered by the RNA-seq analysis. Here we instead focused on assessing the recovery rate of the causal genes of the diagnosed patients.

Q11

Also, please explain how one sample has 293 significant WES variants? The latter sample, 33281, was also one of the outliers in the analysis for MAE....It suggests that this sample does not meet the quality specs, or if so, that the metrics used may leave sample(s) that should be removed?

A: This sample indeed is the one with most new variants when compared to ExAC, indicating a different ethnicity. The observation that the two samples with the highest number of new variants are among the 4 samples that were spotted by the reviewer to have many SNVs with MAE, confirms the quality of the applied analysis. High numbers of (potentially rare) SNVs increase the chance to identify MAE. The number of genes with MAE in patient #33281 (n=11) is not impractically large.

Q12

The authors comment on the paper by Cummings et al. in their discussion. They correctly report that there is a discrepancy in '% patients solved' in both studies (10% here compared to 35% by Cummings et al.). Please discuss why these are so different! The authors do comment that the fact that both studies successfully identify diagnosis shows the value of RNAseq, but this obvious conclusion must be complemented by discussion on the rationale for the differences in percentage diagnosis.

A: Cummings et al. look at the group of neuromuscular disorders. In comparison to mitochondrial disorders, neuromuscular disorders are genetically and clinically better defined. This facilitates the diagnosis. Furthermore, Cummings et al. perform the RNA-seq directly in the affected tissue which has the advantage of circumventing issues related to tissue specific expression. In our case some patients do indeed also have a muscle phenotype, most of our patient present however with either a predominant heart, brain, or liver phenotype. Because of the limited accessibility of these tissues, we wanted to evaluate the applicability of RNA-seq in fibroblasts, as a less invasive biopsy type. We have added those two points to the discussion.

Q13

Figure 1 contains a circle with two shades of blue for 'diagnosis' and 'no diagnosis' on the top, and in addition to those colours, a shaded version on the bottom. The shaded part is likely to represent the 10% diagnoses? But cartoon shows shading that is more than 10%. I also believe

that perhaps these pie charts are better to be included in the last figure, where the summarize the results.

A: This cartoon was indeed meant to be conceptual. We however updated Fig1 so that the dashed lines correspond to 10%. We keep this cartoon in Figure 1 because it gives a clear overview of the strategy and goal of the study.

Q14

I cannot help to be left with the feeling that some of the findings presented (such as the ALDH18A1 and MCOLN1) WES already pointed towards the molecular diagnosis, and that RNAseq was confirming this diagnosis, rather than identifying it. That is, if the molecular diagnostic lab performing WES would have ask for routine RNA work-up of these genes, the diagnosis would have also been performed, without the need of 'full RNAseq'.

A: Given a clear candidate by WES targeted RNA sequencing may be an option. We will describe in the following paragraphs, why the WES variants for ALDH18A1 and MCOLN1 were not prioritized.

Indeed, the variants in ALDH18A1 were detected by WES. The variants (one loss-of-function and one VUS) were even reported to the clinician. But due to an at that time incoherent clinical phenotype of the patient, ALDH18A1 was rejected as the diagnosis (like reviewer 1 who questioned this diagnosis). We were startled when ALDH18A1 popped up again in our RNA-seq analysis and we included the sample in our validation experiments. Quantitative proteomics provided additional evidence which prompted us to perform additional metabolomic studies and supplementation assays. And even though this strengthened our initial assumption, final confirmation only came by a publication of independent individuals with ALDH18A1 resembling the phenotype in our patient. We have now clarified the ALDH18A1 paragraph to make this point clearer.

For MCOLN1, the patient was clinically diagnosed in a clinic in Paris. Biochemical assays evaluating a possible Mucopolidosis could not confirm the clinical diagnosis. Additional tests revealed pathological observation pointing at a mitochondrial disorder. Our WES analysis prioritized coding and direct or near splice variants. Since one variant in MCOLN1 was intronic and not a near splice variant and furthermore in a gene not associated with mitochondrial disorder, MCOLN1 was missed by variant prioritization. Such constellations show the power of an unbiased RNAseq analysis which disclosed the limits of the enzymatic evaluations which missed Mucopolidosis due to MCOLN1 mutations as well as the limits in the interpretation of intronic variants. We had changed the text on MCOLN1 in the last iteration explaining the inconclusive WES analysis in this sample. We do not feel it needs further edition. We would be happy to make changes if this is still unclear.

Finally, we would like to reiterate the argument that full RNA-seq is becoming routine, like WES compared to targeted DNA sequencing and provides more information than a targeted assay.

Point-by-point response to reviews

Reviewer #3 (Remarks to the Author)

I wish to start with an apology for not meeting the deadline for completing this review. But mostly, I would like to thank the authors for providing the detailed responses to my queries. They have been addressed appropriately and I have not further comments for review.

A

We are happy that we could address all the questions of the reviewer.